# Anti-Inflammatory and Antimicrobial Properties of Thyme Oil and Its Main Constituents

**DOI:** 10.3390/ijms24086936

**Published:** 2023-04-08

**Authors:** Evros Vassiliou, Oreoluwa Awoleye, Amanda Davis, Sasmita Mishra

**Affiliations:** Department of Biological Sciences, Kean University, Union, NJ 07083, USA

**Keywords:** thyme oil, carvacrol, thymol, γ-terpinene, p-cymene, linalool, inflammation, cancer

## Abstract

Thyme oil (TO) is derived from the flowers of various plants belonging to the genus *Thymus*. It has been used as a therapeutic agent since ancient times. *Thymus* comprises numerous molecular species exhibiting diverse therapeutic properties that are dependent on their biologically active concentrations in the extracted oil. It is therefore not surprising that oils extracted from different thyme plants present different therapeutic properties. Furthermore, the phenophase of the same plant species has been shown to yield different anti-inflammatory properties. Given the proven efficacy of TO and the diversity of its constituents, a better understanding of the interactions of the various components is warranted. The aim of this review is to gather the latest research findings regarding TO and its components with respect to their immunomodulatory properties. An optimization of the various components has the potential to yield more effective thyme formulations with increased potency.

## 1. Introduction

Thymus is a perennial evergreen herb of the angiosperm plant family Lamiaceae (mint plant family) that has 350 species and 36 subspecies and is native to Europe, North Africa, and Asia [1]. Because of its distinct aroma, the plant is a popular culinary herb. Wild thyme (*Thymus serphyllum*) is the wild relative of all cultivated species [2]. Garden thyme is the most popular and commonly utilized plant for TO (*Thymus vulgaris* L.). Leaf morphology is one of the distinct features unique to the genus Thymus. The leaves are typically tiny (less than 1/8 inch long and 1/16 inch wide), narrow and elliptical, greenish-gray in color, and grouped in whorled phyllotaxy. Thyme flowers are white, yellow, or purple whorls that terminate branches. Some of the widely used cultivars of thymes for TO are *Thymus rumidicus hispánicos*, *Thymus zygis*, *Thymus vulgaris*, *Thymus hyemalis*, *Thymus mastichina*, *Thymus citríodotus*, *Thymus corydothymus*, *Thymus loscossi*, *Thymus pipirella*, and *Thymus communis*.

Thyme oil (TO) is a complex mixture of at least 70 different species [3]. Thymol tends to be among the most prevalent molecular species in TO in species such as *Thymus vulgaris* and *Thymus magnus* Nakai (Table 1). It is also the one that has been studied the most and has been shown to exert a range of therapeutic properties that includes antimicrobial, antitumoral, antifungal, antiparasitic, antioxidative and anti-inflammatory. P-cymene is another compound present in TO at relatively high concentrations with antiviral, antioxidant, and antitumoral properties particularly when conjugated to Ruthenium [4,5,6,7]. γ-Terpinene is another prevalent constituent in TO that belongs to a group of compounds called monoterpenes and has been shown to inhibit growth of helminthic and protozoan infections but only as a component in Australian tea tree oil [8]. Carvacrol, which is structurally related to thymol, has potent antimicrobial and antifungal properties [9,10]. Besides TO, carvacrol is present in essential oils derived from oregano, pepperwort, and wild bergamot plants. Its relative abundance can vary significantly depending on the source. (E)-β-Caryophellene is also a constituent of TO with anti-inflammatory and proapoptotic properties in tumor cells [11]. Consistent with thymol and carvacrol, (E)-β-Caryophellene exhibits antibacterial growth properties [12]. Linalool, a terpene alcohol also found in flowers and spices, is another TO component. Linalool exhibits antioxidant properties and reverses oxidative. In addition, linalool exhibits proapoptotic properties in cancer cells similar to other components of TO [13].

## 2. Thymol

Thymol is one of the main constituents of TO (see Figure 1 for common TO constituents). Depending on the species (over 350 species are known per POWO), levels of thymol in the extracted oil can vary considerably. Thymol exhibits a variety of therapeutic properties and is among the most investigated components of TO.

Multiple studies highlight the anti-inflammatory properties of thymol [18,19,20]. In a study utilizing *Staphyloccocus aureus* membrane vesicles and keratinocytes, thymol reduced expression levels of IL-1β, IL-6, TNF-α, IL-8, and MCP-1 [19]. The anti-inflammatory effect was observed in a mouse in vivo system as well, where thymol suppressed Th1-, Th2-, and Th17-mediated inflammatory responses. Using the human peritoneal mesothelial cell line HMrSV5, Wang Q et al. (2018) showed that thymol reduced phosphorylation of multiple mediators involved in the NFκB inflammatory pathway [18]. Thymol reduced phosphorylation of IKK, IκΒα, and ΝFκΒp65 and resulted in reduction of MCP-1, TNF-α, and IL-6. The anti-inflammatory outcome of thymol was investigated in a human clinical trial involving gingival inflammation [20]. Varnish containing thymol/chlorhexidine was used in 25 individuals with fixed orthodontic appliances and gingival inflammation. The thymol/chlorhexidine varnish reduced levels of Prostaglandin E_2_ (PGE_2_) within 8 days. In a different study, a combination of thymol and carvacrol reduced production of the inflammatory cytokines IL-25, IL-33, and Thymic Stromal Lymphopoietin (TSLP) in BEAS-2B-transformed human bronchial epithelial cells stimulated with chitin [21]. SHIP-1 and SOCS-1, known downregulators of TLR-2 and TLR-4 inflammatory signaling, were elevated with thymol/carvacrol.

A recent study utilizing U87 human malignant glioblastoma cells has shown that thymol induces apoptosis, increases Reactive Oxygen Species (ROS), and increases levels of the proapoptotic factors Bax and p53 [22]. Similar proapoptotic effects were observed in bladder cancer cells by activating the apoptotic intrinsic pathway via caspase-3, caspase-9, release of cytochrome c, and reduction of the antiapoptotic protein bcl-2 [23]. Thymol was effective in reducing cell proliferation and inducing apoptosis of colorectal cancer cells [24]. The antitumoral properties of thymol were linked to the activation of the Bax/bcl-2 pathway. Similar findings were seen using the acute promyelotic cancer cell line HL-60 [25]. As with previous studies, an increase in caspase-3, 8, and 9 was observed accompanied with an increase in Bax and a reduction in bcl-2.

Thymol has also been evaluated for its potential application as an antimicrobial agent. A thymol concentration of 100 μg/mL reduced viability of the dental caries causing bacteria *Streptococcus mutans* by 50% [26]. Thymol can inhibit the growth of *S. mutans* by inducing autolysis, stress growth inhibition, and reduced biofilm formation. Currently, thymol-rich extracts are generally used as an expectorant in coughs associated with cold and also in dentistry as a disinfectant [27]. Thymol is effective against both gram-positive and gram-negative bacteria [28]. The study of Thoshar et al. (2013) tested different essential oils rich in phenols and thyme oil for minimal inhibitory concentration (MIC) and determined that 2 µL/mL was effective against *E. coli*. However, for *E. faecalis* and *S. aureus*, growth inhibition required a higher concentration. Antibacterial properties of thyme oil rich in thymol was also tested against the methicillin resistant (MRSA) strain [29,30]. Kryvtsova et al. (2019) tested thyme oil extracted from *T. vulgaris* to study the antimicrobial properties using MRSA strain *S. aureus* isolated from the oral cavity of patients suffering from periodontitis and pharyngitis. Thyme oil concentration of 0.01% (*v*/*v*) and 0.05% effectively reduced the growth up to 53% and 76%, respectively. The main components of volatile oil used in this research were phenolic monoterpenes, including thymol. In a similar study by Tohidpour et al. (2010), thymol extracted from *T. vulgaris* was more effective than essential oil from *Eucalyptus globulus* Labill in controlling the growth of MRSA [31]. Surprisingly, the antibacterial properties of TO also depend on different growth ages of thyme plants [3]. For example, thyme oil before flowering and after flowering has significant differences in controlling the growth of *Pseudomonas aeruginosa* even though the thymol level difference was nonsignificant (55.81% and 54.21%, respectively). Recently, thymol-rich Thyme species are gaining attention for their antifungal properties. For example, studies on antifungal properties of thyme oil extract rich in thymol has been shown to be effective against growth of the fungal genus *Cryptococcus*, which is a pathogen responsible for cryptococcosis [32]. In this study, among six tested compounds, the combination of thymol and carvacrol was effective four to eight times higher than the standard therapeutic drugs. The potential mode of action of thymol against fungi is based on the target-specific effects of thymol against ergosterol, which is the unique sterol specific to fungal cells [27].

Recently, Liggri et al. (2023) have shown that both thymol and carvacrol bind to the novel mosquito protein AgamOBP5, an odorant-binding protein that is involved in the odorant perception process of mosquitoes [33]. This was the first study to show the binding of a natural plant-derived insect repellent to an odorant-binding protein. Furthermore, the study provides a mechanistic explanation for the insect repellent properties of TO and specifically thymol and carvacrol.

## 3. Carvacrol

Carvacrol content in TO also varies significantly depending on the thyme species. TO derived from *Thymus capitatus* plants was shown to be very high in carvacrol (79.9%) [34]. On the other hand, TO derived from *Thymus vulgaris* plants was shown to be much lower in carvacrol (1.56%) [3]. Similar to thymol, carvacrol exhibits anti-inflammatory properties and has been shown to protect retinal pigment epithelial cells against inflammation, oxidative stress, and apoptosis induced by high glucose levels [35]. Mechanistically, the transient potential melastatin 2 (TRPM2) cation channel was inhibited by carvacrol and thus provided protection against excessive calcium influx and ROS species. In addition, Glutathione and Glutathione Peroxidase were elevated with carvacrol, providing protection against the ROS species generated from high glucose levels. Similar anti-inflammatory findings were seen with human tonsil epithelial cells stimulated with Lipoteichoic Acid and Peptidoglycan [36]. IL-6, IL-8, ENA-78, and GCP-2 were all suppressed with carvacrol. Furthermore, PGE_2_ and COX-2 levels were reduced with carvacrol.

Carvacrol showed analogous antitumoral properties as thymol in a study involving oral squamous carcinoma cells [37]. Keap1/Nrf2 and NALP3 were upregulated in the carcinoma cells. Treatment with carvacrol inhibited both Keap1/Nrf2 and the NALP3 proteins and resulted in suppression of clone formation and migration capacity. Antiproliferative effects of carvacrol were observed in a study with osteosarcoma cells [38]. Carvacrol induced apoptosis in U2OS and 143B cells. Similar to thymol, a reduction in bcl-2 and an increase in Bax were detected after carvacrol treatment. A reduction in migration and invasion of the U2OS and 143B cells was correlated with a reduction in MMP-9 expression. Treatment with carvacrol showed similar anticancer effects in a study with breast cancer cells [39]. The effect was related to carvacrol’s ability to inhibit the transient receptor potential melastatin-like 7 channel (TRPM7), which regulates the cell cycle of cells. At 200 μΜ, carvacrol increased the number of cells in the G1/G0 phase and decreased the number of cells in the S and G2/M phases. Treatment with carvacrol reduced tumor size in mice transplanted with glioblastoma cancer cells [40]. Carvacrol’s inhibition of TRPM2 and TRPV4 is recognized in several studies [41]. Its ability to inhibit these channels makes it suitable as an anticancer agent and an antioxidative agent. Elevation of Glutathione and Glutathione Peroxidase in the presence of carvacrol in conjunction with the inhibition of the TRPM2 and TRPV4 channels act together to confer cellular protection against oxidative stress. The antioxidative nature of carvacrol and its therapeutic effects have been demonstrated in a mouse in vivo model of Alzheimer’s disease [42]. Intraperitoneal administration of carvacrol resulted in increased cellular viability in the experimental animal model of Alzheimer’s disease that uses Aβ1-42 bilateral intrahippocampal injection. Memory impairment assessed with a passive avoidance test was improved, providing support for carvacrol as a promising agent in neurodegenerative diseases. In a human clinical trial involving moderately asthmatic patients, carvacrol was effective in reducing the levels of inflammatory cytokines and oxidative stress markers and shown to improve pulmonary function tests [43]. The study was a randomized, placebo-controlled, double blind study involving 33 moderately asthmatic patients. The carvacrol-treated group received 1.2 mg/kg/day for a total of 2 months. Improvement of respiratory symptoms were observed by the first month, providing further support that carvacrol is a natural antioxidant and anti-inflammatory agent with therapeutic benefits in several diseases. 

In addition to its anti-inflammatory properties, carvacrol has been extensively studied for its antibacterial properties. Several studies investigating the effects of carvacrol extracted from different Thyme species on various bacterial strains have been carried out. Carvacrol extracted from *Thymus vulgaris* L. has been shown to exhibit potent antibacterial activity against several strains of bacteria, including *Escherichia coli*, *Staphylococcus aureus*, and *Pseudomonas aeruginosa* [44]. In their study, Soković et al. (2010) reported that carvacrol extracted from *T. vulgaris* L. had a minimum inhibitory concentration (MIC) of 0.5–1.5 mg/mL against these bacterial strains. Similar results have been observed by Gedikoğlu et al. (2019) in evaluating the antimicrobial properties of *T. vulgaris* L. on a series of gram-positive and gram-negative bacteria (*Bacillus cereus*, *Staphylococcus aureus*, *Staphylococcus epidermidis*, *Escherichia coli*, *Salmonella enteritidis*, and *Salmonella typhimurium*) [45]. Surprisingly, extraction methods might affect the antibacterial properties of bioactive compounds. TO extracted by microwave-assisted extraction displayed significantly (*p* < 0.05) higher antibacterial activity against bacteria than did the hydrodistilled essential oil extraction. Antibiotic-resistant pathogenic bacteria are gaining more attention and various plant derived species are being explored as alternative antimicrobial agents. A comparative study on essential oil extract from different herbs including *T*. *vulgaris* reported the synergistic effectiveness of carvacrol against 32 erythromycin-resistant Group A *Streptococci* [46]. The Minimum Inhibitory Concentration (MIC) of carvacrol was found to be 64 μg/mL, which was effective against inhibiting growth Group A *Streptococci*. The same study evaluated the antibacterial properties of carvacrol alone and in combination with erythromycin. Surprisingly, carvacrol from *T*. *vulgaris* alone was effective in both treatment types’ ability to inhibit growth of Group A *Streptococci*, further enhancing the potential use of carvacrol as an effective phytotherapeutic agent against antibiotic-resistant bacteria [46]. Carvacrol from other Thyme species was also evaluated to determine the effectiveness of antibacterial properties.

Among different chemotypes, hydrocarbon monoterpenes in essential oils extracted from *Thymus* spp. show the lowest antibacterial activity compared to oxygenated compounds, especially phenol-type compounds such as thymol and carvacrol [44]. The antibacterial properties of carvacrol extract from other *Thyme* spp. were also evaluated. Carvacrol extracted from *T*. *zygis* L. was found to be effective against several bacterial strains, including *S*. *aureus* and *E*. *coli*. An MIC of 0.16–0.63 mg/mL carvacrol extracted from *T*. *zygis* L. was found to be effective against controlling the growth of these bacterial strains. The antimicrobial properties of essential oil from the genus *Thymus* could be explained by their high percentage of phenol components. Carvacrol might interfere with the activity of cell wall enzymes, thus inhibiting bacterial growth by damaging their cell wall. A recent report investigating the antibacterial properties of carvacrol provided the possible mechanism of carvacrol against *Streptococcus pyogenes* [47]. Carvacrol exhibits rapid antibactericidal activities by inducing morphological changes, cytoplasmic leakage, and, consequently, cell damage.

The potential application of carvacrol from various *Thymus* cultivars as antifungal agents has been reported recently [10,48,49,50]. Boukhatem et al. (2020) reported the antifungal activity of essential oil extracted from *T*. *vulgaris* against eight yeast species and eight filamentous fungi isolated from mucocutaneous fungal infection pathogens for skin diseases. For this study, TO was extracted from fresh leaves and floral heads. Out of 25 bioactive compounds in the thyme oil, carvacrol (56.8%) was detected in high amounts, followed by p-cymene (12.8%), γ-terpinene (11.17%), and thymol (3.99%). A series of fungal species were used for the sensitivity assay, which included different strains of yeast (*Candida albicans*, *C*. *tropicalis*, *C*. *parapsilosis*, *Trichosporon*, and *Rhodotorula* sp.), and filamentous fungus (*Aspergillus terreus*, *A*. *flavus*, *A*. *niger*, *A*. *fumigatus*, *Mucor*, and *Penicillium*). Out of three concentrations (20, 40 and 60 μL/disc), TO concentration of 60 μL was effective against all tested yeast strains as compared to the positive control (1% Hexamidine). *Candida tropicalis* and *C*. *parapsilosis* were the most vulnerable strains to the TO, with zones of inhibition of 50 mm and 60 mm. Among filamentous fungi, TO was more effective against *Aspergillus terreus* and *A*. *fumigatus* (zones of inhibition 55 and 45 mm, respectively). Similar antifungal properties of carvacrol against *Candida tropicalis* were reported by other studies as well [51,52]. In addition, a number of studies have confirmed the antifungal properties of essential oils and extracts from different varieties of the genus *Thymus* rich in monoterpenic alcohols and/or volatile phenols [53,54]. Another application of thyme oil rich in carvacrol acting as an antifungal agent has been reported in controlling *Botrytis cinerea,* a primary pathogen causing stem and fruit rot during pre- and postharvest of ornamental crops, fruits, and vegetables [10]. Out of several concentrations tested at 140 µL/L, carvacrol was effective in disrupting the mycelia. A possible mechanism of carvacrol-rich TO extract is a drop in extracellular pH, thus inducing membrane damage and a decrease in cellular lipid content. Other *Thymus* species such as *T*. *zygis* and *T*. *vulgaris*, with higher quantities of phenols, also demonstrated a wide spectrum of inhibitory growth against a range of pathogenic filamentous fungi and yeasts, with decreased sensibility to antifungal drug [53,54,55]. However, carvacrol was more active against dermatophyte strains, in a similar manner to the extracted oil.

## 4. P-Cymene

P-cymene is typically found at lower concentrations in TO compared to thymol and is a component of other essential oils including cinnamon and cumin [56]. P-cymene has an overall anti-inflammatory character and has been shown to reduce allergy-associated immunological markers in both human and murine allergy models [57]. Treatment of human monocyte-derived mature dendritic cells from grass or birch pollen allergic donors with cinnamon extract, p-cymene, or trans cinnamaldehyde inhibited their maturation. When cocultured with autologous CD4+ in the presence of specific antigen-pulsed dendritic cells, CD4+ proliferation and Th1/Th2 cytokine production was inhibited. P-cymene showed similar characteristics by inhibiting IL-8 release in LPS-stimulated THP-1 monocytes [58]. Synergism between p-cymene and trans cinnamaldehyde has been observed. Reduction in phosphorylation of Akt and IκB seem to play a key role in the observed anti-inflammatory effects. In the colitis rat model induced with trinitrobenzene sulphonic acid (TNBS), p-cymene with rosmarinic acid reduced levels of malondialdehyde (MDA) and myeloperoxidase while restoring glutathione levels [59]. Levels of IL-1β and TNF-α were also reduced. COX-2 levels in spleen, mesenteric lymph node, and colon samples declined as well. Overall, p-cymene and rosmarinic acid provided intestinal cytoprotection and maintained the mucous layer.

P-cymene’s antitumoral effects were reported in a high-fat diet colorectal cancer rat model [60]. A significant reduction in cancerous nodules in the p-cymene group was observed. Leptin and IL-1 levels were decreased, but interestingly, IL-6 levels were increased. In a similar colorectal cancer study involving the hyperlipidemia/dimethyl hydrazine rat model, p-cymene’s anticancer properties were linked to its antioxidant and anti-inflammatory properties [61]. Reduction in IL-6, COX-2, IL-1, and adiponectin was observed. Similar reduction in the oxidative markers Superoxide Dismutase and Malondialdehyde was detected. In addition to being utilized alone as an anticancer molecule, p-cymene has been used as a ligand for the metal Ruthenium, which gives rise to a number of organometallic compounds with promising anticancer properties [5]. Besides its antitumoral properties, p-cymene exhibits antinociceptive effects [62]. A reduction in mechanical hyperalgesia, spontaneous nociception, and nociception induced by nonnoxious palpation was achieved with p-cymene. Calcium channel current modulation was proposed as a mechanism for the antinociception.

## 5. γ-Terpinene

γ-Terpinene belongs to a group of isomeric compounds referred to as monoterpenes. Their difference lies in the position of the two carbon–carbon double bonds (see Figure 1). γ-Terpinene is the precursor molecule for both thymol and carvacrol. The structure and characteristics of the enzyme responsible for its synthesis from geranyl diphosphate, γ-terpinene synthase, has been elucidated and studied in detail [63]. In TO, γ-Terpinene is the most prevalent among α, β, and δ isomers. Species such as *Thymus vulgaris* have a much lower content in relation to the *Thymus capitatus* species [3,34]. It is unclear if the variation in γ-Terpinene translates to different therapeutic properties.

There is much less research on γ-Terpinene compared to thymol, carvacrol, and p-cymene. The antimicrobial properties of *Thymbra capitata* TO and, by extension, its components were mostly attributed to p-cymene and carvacrol where strong synergism was reported against Gardnerella spp [64]. Nonetheless, a number of studies have shown that γ-terpinene exhibits anti-inflammatory characteristics similar to carvacrol and thymol but through a different mechanism. Ramalho et al. (2016) showed that murine LPS-stimulated peritoneal macrophages produced less IL-1β and IL-6 when treated with γ-Terpinene but more of the anti-inflammatory cytokine IL-10 [65]. Interestingly, the increase in IL-10 was due to the elevation of the COX-2 enzyme and its downstream arachidonic acid metabolite PGE_2_. Inhibition of the COX-2 enzyme with nimeluside (selective COX-2 inhibitor) abolished the IL-10 increase. Furthermore, with diminished IL-10 production, the inhibition of IL-1β and IL-6 was lost. γ-Terpinee in IL-10-deficient mice did not produce any inhibition on IL-1β and IL-6, suggesting the PGE2/IL-10 pathway as a mechanism for its anti-inflammation. This pathway is not utilized by carvacrol, thymol, and p-cymene and makes γ-terpinene unique particularly in terms of the induction of IL-10.

## 6. Linalool

Linalool is another constituent of TO, but with lower prevalence than thymol, carvacrol, and p-cymene. A terpene alcohol, linalool is found in a variety of flowers and seeds and their essential oils [66]. Linalool has a pleasant smell and is widely used in the fragrance industry [67].

Multiple studies have investigated linalool’s anti-inflammatory qualities in a variety of systems. Cigarette smoke-induced inflammation was inhibited by linalool in mice [68]. Reduction in levels of the inflammatory cytokines/chemokines TNF-α, IL-6, IL-8, IL-1β, and MCP-1 were detected by ELISA. The inhibition was mediated via the NF-κB pathway. Similar findings were reported in a study involving LPS-stimulated RAW 264.7 cells and LPS-induced in vivo lung injury mouse models [69]. Analogous to the cigarette smoke-induced inflammation study, levels of TNF-α and IL-6 were reduced with linalool. The investigators showed that phosphorylation of IkBα was reduced along with p38 and c-Jun.

Protective effects of linalool have been reported in neurodegenerative animal models as well [70]. The animals used were a triple transgenic Alzheimer’s disease model (3xTg-AD) in which the linalool-treated group (25 mg/kg, every 48 h, 3 months) showed a reduction in the inflammatory markers p38 MAPK, NOS2, COX-2, and IL-1β in the hippocampi and amygdalae. Reduction in extracellular β-amyloidosis, tauopathy, astrogliosis, and microgliosis was accompanied by improved learning and spatial memory in the elevated plus maze test. Linalool’s anti-inflammatory properties were also observed in mice with allergic asthma [71]. A decrease in iNOS expression and PKB activation in lung tissues occurred with linalool. Similar to previous findings, downregulation of NFκB and MAPK and diminished signaling were responsible for the anti-inflammation. In another study involving streptozotocin-induced diabetic rats, linalool provided renoprotection [72]. Expression of NF-κB and TGF-β1 was reduced in the kidneys, and an overall reduction in diabetes-induced nephropathy was observed. Anti-inflammatory effects with linalool were reported in a UVB acute radiation-induced inflammation mouse skin model [73]. Topical application of linalool or intraperitoneal injection reversed the expression of COX-2 and ornithine decarboxylase. Furthermore, lipid peroxidation and antioxidant depletion were reduced. In the chronic UVB radiation model, there was a reduction in the classical inflammatory mediators NF-κΒ, TNF-α, IL-6, and COX-2 along with a reduction in tumor incidence. Tumor development was confirmed histopathologically in terms of dysplasia and squamous cell carcinoma presence. Related to its anti-inflammatory nature, linalool behaves as an antioxidant. In the neuronal HT-22 cell line, glutamate-induced mitochondrial oxidative stress was reduced in the presence of 100 μM linalool [74]. Mitochondrial membrane potential (ΔΨ) preservation accompanied by a reduction in mitochondrial ROS and calcium levels translated into increased neuronal survival during glutamate-induced toxicity.

## 7. Discussion

Thyme oil (TO) has a proven record of therapeutic properties and characterized in detail anti-inflammatory mechanisms (see Figure 2). However, it is composed of a very heterogenious mixture of a large number of molecular species. Adding to its heterogeneity is the large number of different plants from which TO can be extracted from. Furthermore, the phenophase of the thyme plant has a direct impact on the concentrations of the various constituents comprising the TO, which can yield different properties. Even the extraction process can yield TO oil with uneven characteristics. Nonetheless, thymol, carvacrol, p-cymene, γ-terpinene, and linalool are very likely among the key components of TO with the most potent therapeutic properties. Given the observed synergism and antagonism that the different molecular species can exhibit, it is rational to consider employing defined formulations of the key components of TO that are likely to be more potent than the natural product. This approach will require testing and validation of concentrations of various species that normally do not exist in the natural product. The potential of harnessing more potent formulations with more consistent therapeutic characteristics is worth the effort.

In addition, future research directions relating to TO and its derivatives may include synergistic formulations with other drugs. For example, the anti-inflammatory properties of thymol, carvacrol, p-cymene, γ-terpinene, and linalool can be used in conjunction with widely used corticosteroids. Formulations composed of synthetic corticosteroids and TO constituents may allow for lower concentrations of the synthetic component and achieve a reduction in side effects typically seen when using only the synthetic drugs. A similar approach may be employed with antibiotics where the use of TO components along with antibiotics may permit a reduction in concentration of the synthetic antibiotic and a reduction in side effects and antibiotic resistance. This strategy has been used for many years now with clavulanic acid and amoxicillin. Another future direction worth investigating is the use of carvacrol and thymol in combination with known synthetic insect repellants where the formulation will permit a lower concentration of the synthetic repellants. The natural scent of carvacrol and thymol will be an added benefit.

In conclusion, there is ample scientific evidence that TO and its constituents have diverse therapeutic properties. However, it is unlikely that they can compete or replace synthetic drugs with similar and often more potent properties. Future research should focus on optimizing formulations of TO constituents by themselves and with synthetic drugs to maximize their potency while reducing their toxicity.

## Figures and Tables

**Figure 1 ijms-24-06936-f001:**
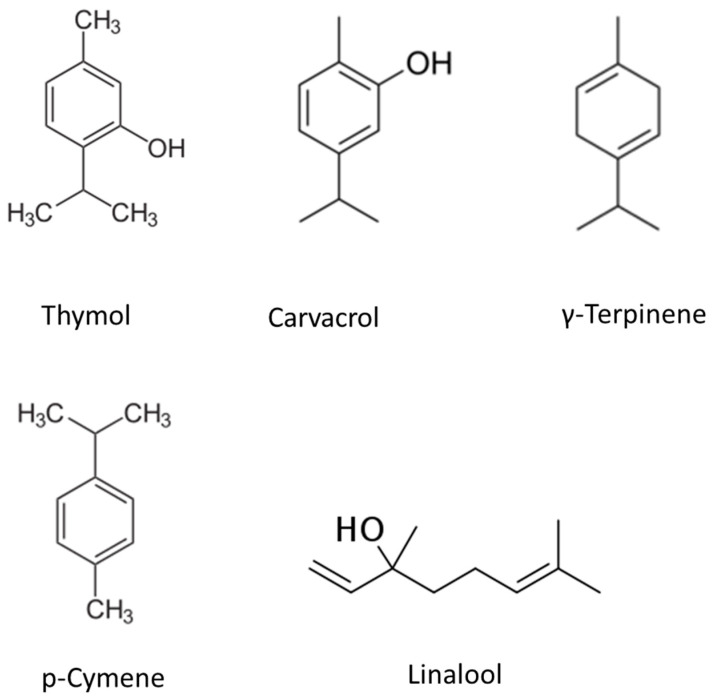
Chemical structures of thymol, carvacrol, p-cymene, γ-terpinene, and linalool.

**Figure 2 ijms-24-06936-f002:**
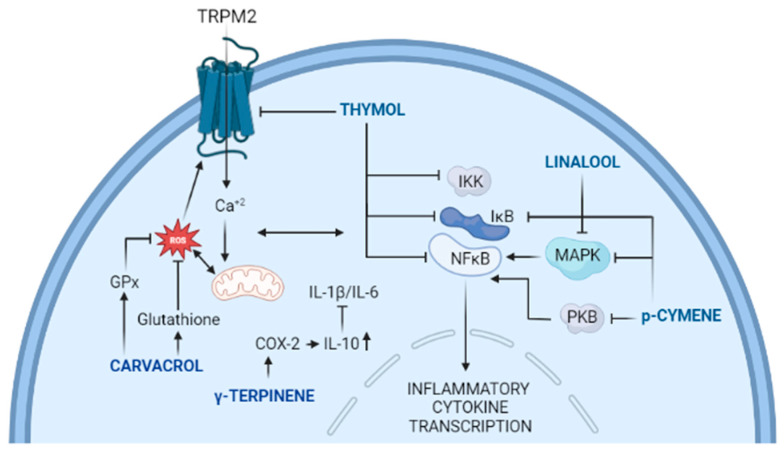
A simplified schematic depiction of the anti-inflammatory pathways utilized by thymol, carvacrol, p-cymene, γ-terpinene, and linalool.

**Table 1 ijms-24-06936-t001:** Prevalence of main thyme oil molecular species in different thyme plants.

*Thymus* spp. *(T.)*	Thymol (%)	Carvacrol (%)	Linalool (%)	γ-Terpinene (%)	P-Cymene (%)	Reference
*T*. *capitatus* (L.) Hofmgg. Link	8–61	14.2–81.2	0–0.4	2.6–33.4	5–22.8	[14]
*T*. *serpyllum*	0.6–0.8	6–20	0.4–63	2.5–4.0	2–9.1	[15]
*T. pulegoides*	0.2–2.2	9.5–18	0.6–13	1.5–8	1.6–6	[15]
*T. marschtallianus*	2–2.2	6.5–9	3.5–4.5	4.0	1.0	[15]
*T. vulgaris* L.	54.2–55.8	2.3–2.9	1.46–2.1	0.82–1.4	12.89–20.6	[3]
*T. atticus*	0.7	0.3	1.0	Trace	0.2	[16]
*T. leucotrichus*	2.7	0.6	1.8	Trace	0.2	[16]
*T. striatus*	1.9	4.3	0.9	Trace	0.2	[16]
*T. perinicus*	20.9	1.1	4.6	0.5	4.8	[16]
*T. zygioides*	51.2	2.9	0.1	1.1	6.5	[16]
*T. quinquecostatus* Celak.	39.8	2.6	0.1	10	9.2	[16]
*T. magnus* Nakai	54.7	3.2	Trace	6.4–15.0	3.5–6.7	[17]

## Data Availability

Not applicable.

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
