# Peer review of "Anti-Inflammatory and Antimicrobial Properties of Thyme Oil and Its Main Constituents"

_ijms, 2023, doi:10.3390/ijms24086936_

Round 1
Reviewer 1 Report
To The Authors,
Vassiliou, E., et al, have composed a comprehensive review on the therapeutic utility of Thyme Oil (TO) and its components on immunomodulation. The review is detailed and the authors have reviewed different chemical components of TO and the known role in immunomodulation so far. The language is easy to follow and there are no major edits to be made.
Author Response
Thank you for your review.
Reviewer 2 Report
Just a minor comment:
Since this review article comes under the Macromolecule section, the authors should cite a macromolecular structure paper in this review article. Overall, the content is good, but it would have been better if a recent PDB structure was added. Reference is as follows:
RCSB PDB - 8BXV: Crystal structure of Odorant Binding Protein 5 from Anopheles gambiae (AgamOBP5) with Thymol
Also, it's advisable to show how the individual Thymol compounds triggers immune response within cell with a couple of figures.
Author Response
Thank you for the suggestion regarding the study that shows the binding of thymol and carvacrol to AgamOBP5.
We included the following the comments along with the reference:
Recently, Liggri et al., (2023) have shown that both thymol and carvacrol bind to the novel mosquito protein AgamOBP5, an odorant binding protein that is involved in the odorant perception process of mosquitoes [33]. This is the first study to show the binding of a natural, plant derived insect repellent to an odorant binding protein. Furthermore, it provides a mechanistic explanation for the insect repellent properties of TO and specifically thymol and carvacrol.
Regarding thymol's anti-inflammatory properties, figure 2 depicts the target proteins i.e NF-κB, IκΒ, ΙΚΚ and TPRM2.
Once again thank you for the positive review.
Evros Vassiliou.